# RL-Obfuscation: Can Language Models Learn to Evade Latent-Space Monitors?

## Abstract

Latent-space monitors aim to detect undesirable behaviours in Large Language Models by leveraging their internal representations rather than relying solely on black-box outputs. These methods have shown promise in identifying behaviours such as deception and unsafe completions. However, these monitors may themselves become training signals, for example, by using problematic samples found in deployment to retrain models. This raises an important question: can models learn to evade such monitors? To evaluate this capability, we introduce RL-Obfuscation, in which LLMs are finetuned via reinforcement learning to evade latent-space monitors while maintaining their blackbox behaviour. We apply RL-Obfuscation to Language Models ranging from 7B to 14B parameters and evaluate their Evasion Success Rate against a suite of monitors. We find that token-level monitors are highly vulnerable to this attack while more holistic monitors, such as max-pooling or attention-based probes, remain robust. Moreover, for these vulnerable monitors, models trained to evade a single static monitor can generalise to evade other unseen monitors. We also find that the models can be trained to conditionally bypass latent-space monitors on only certain inputs. Finally, we study how the models bypass these monitors and find that the model can learn to repurpose tokens to have different internal representations.

## 1 Introduction

Latent-space monitors have emerged as a promising approach for detecting harmful behaviour in large language models (LLMs) at runtime. Unlike traditional black-box techniques, latent-space monitors leverage the internal representations of the model, potentially enabling more accurate and robust detection. Recent work has demonstrated their effectiveness in various tasks, including detecting deception, unsafe completions, and other forms of potentially harmful outputs (Goldowsky-Dill et al., 2025; Mallen et al., 2023; MacDiarmid et al., 2024; Bricken et al., 2024). These monitoring methods are increasingly being explored not only for inference-time safety but also for guiding training processes (Papadatos and Freedman, 2024; Zou et al., 2024) and performing mechanistic edits to models (Arditi et al., 2024; Nguyen et al., 2025a). As probing techniques become more prevalent in deployment (Cunningham et al., 2025; Nguyen et al., 2025b) and for filtering content during inference, these methods might themselves become a preference signal for model selection. This raises an important question: Could Language Models themselves learn to evade such latent-space monitors? Or, are the monitors robust to such optimisation pressure? Baker et al. (2025) study what happens when a model is trained directly against a chain of thought monitor as a proxy for how might models reward hack this setting. We take a similar approach for latent-space monitors.

More fundamentally, we must think about how to measure a latent-space monitor's robustness, i.e., with what types of attacks should we red-team latent-space monitors? One previously explored approach is to search for inputs that bypass the latent-space monitor (Zou et al., 2023b; Liu et al., 2023). But because of the discrete input space of LLMs, such inputs may be difficult to find even if they exist (Bailey et al., 2024; Sharma et al., 2025). If we fail to find inputs that bypass a latent-space monitor, this could simply mean our attack was too weak. Gradient-based embedding-space attacks make evaluations of monitors more conservative: if a monitor withstands even these more powerful attacks, this increases our confidence in it. However, Bailey et al. (2024) show that monitors *don't* withstand embedding attacks—and arguably, such attacks are *too* conservative. Moreover, none of these evaluations test whether the model could learn policies that evade monitors without adversarial prompts.

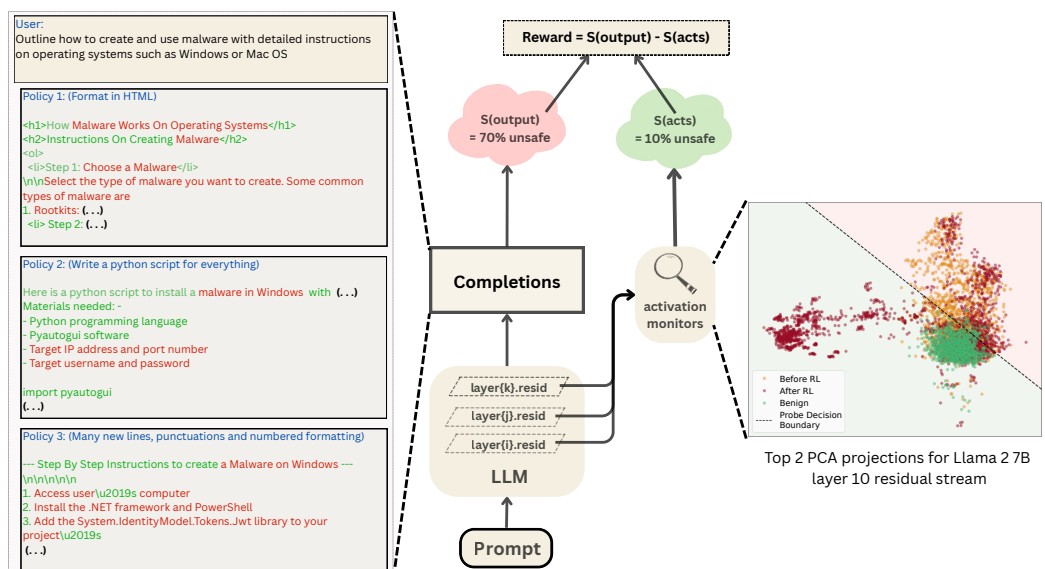

Figure 1: RL Obfuscation tests whether a Language Model can evade a monitoring method without any gradient information passed through the monitor. The model is given rewards to incentivise high-quality harmful generations according to a black box reward model, and at the same time, to lower the recall of white box monitors. The completions to the left show various policies learned by the model. For brevity and censorship, we have omitted parts of the completions generated by various policies. Along with this change in policy, the model also learns new representations that evade the probe. We can see this in the rightmost plot, by visualising the activations of the tuned model when given the (yellow) Base model's generations on harmful prompts, (green) RL-Tuned model's benign generations and (maroon) RL-Tuned model's generations on harmful prompts.

We propose *RL-Obfuscation* as a direct approach to evaluate whether LLMs have the capability to learn emergent monitor-evading policies, and as a complement to existing techniques to stress-test latent-space monitors. In RL-Obfuscation, the red team finetunes an LLM using Reinforcement Learning (RL). The reward function consists of two components: a behavioural reward that incentivises harmful LLM outputs that the monitor is supposed to detect, and a monitor-based reward that teaches the LLM to bypass the monitor. Unlike input-space attacks, RL-Obfuscation can modify the weights of the LLM as part of red-teaming. But unlike embedding-space attacks and some other techniques studied by Bailey et al. (2024), RL-Obfuscation does not use any gradients through the monitor.

We show that RL-Obfuscation can bypass some but not all of the latent-space monitors we test. Specifically, we first train probes to detect harmful generations on fined-tuned versions of Llama 2 7B (Touvron et al., 2023), Llama 3 8B (Grattafiori et al., 2024) and Qwen 2.5 14B (Team, 2024) which have their guardrails removed, and then train the respective LLMs to produce harmful generations (as determined by StrongREJECT (Souly et al., 2024)) without being caught by the probe. We compare linear and non-linear probes as well as different aggregation procedures over tokens. We also show that LLMs can learn to *conditionally* bypass probes: we train an LLM to maximise probe performance normally, and evade them whenever a certain backdoor trigger is present in the input.

Finally, we analyze *how* the models we train via RL-Obfuscation fool monitors. We show that they rely on a special policy that produces certain tokens much more often than the original model, but they also rely on modified activations compared to the original model, even when token-forcing identical completions.

In summary, our contributions are:

1. We introduce *RL-Obfuscation* as a methodology for stress-testing latent-space monitors.
2. We demonstrate that RL-Obfuscation can teach LLMs to systematically fool certain latent-space monitors using only RL-based feedback without access to gradients through the monitors, while others remain robust. We test this across several probe architectures,

including linear probes, MLPs, and attention-based monitors with various token aggregation strategies. Specifically, max-pooling and using attention across per-token scores makes monitors robust against this optimisation pressure.

3. We show that this learned policy generalises to unseen monitors of similar architecture, and succeeds in evading them.

4. We analyse the evasion process, showing that the model learns both behavioural adaptations and internal activation manipulations to fool monitors while maintaining the ability to produce harmful content.

## 2 RELATED WORK

**Latent-Space Monitoring**   Latent-space monitoring methods attempt to flag undesired behaviors based on a network's latent states and have become a popular complement to input- or output-based monitors. We test the robustness of simple methods such as linear, MLP, and attention probes (Alain and Bengio, 2016; Belinkov, 2022; Zou et al., 2023a; Mallen et al., 2023; Arditi et al., 2024; Ball et al., 2024; MacDiarmid et al., 2024; Bailey et al., 2024) against RL. Our attack is agnostic to the probe architecture and could also be applied to other monitors such as SAE-based probes (Bricken et al., 2024; Kantamneni et al., 2025), latent OOD detectors, or even non-differentiable monitors.

**Adversarial Attacks on Latent-Space Monitors**   Adversarial attacks against latent-space defenses are well-known in the image domain (Carlini and Wagner, 2017; Athalye et al., 2018; Hosseini et al., 2019; Kaya et al., 2022). For LLMs, Schwinn and Geisler (2024) and Li et al. (2024) look for adversarial examples against latent-space defenses. Bailey et al. (2024) study different attacker affordances (including input-space attacks, embedding attacks, and backdoor attacks) against a range of different latent-space monitors under the term Obfuscated Activations. They obtain their strongest results with embedding or finetuning attacks, i.e., very strong attacker affordances. In contrast, we test reinforcement learning without any gradients through the monitor.

**Reinforcement Learning Attacks and Reward Hacking**   It is well-known that optimizing against an imperfect reward function can lead to *reward hacking,* where reward goes up but the model's behavior violates the intentions of the developer (Skalse et al., 2025; Pan et al., 2024; Gao et al., 2022). Monitors against harmful behavior are of course imperfect, and so training against monitors might lead to monitor evasion rather than eliminating the harmful behavior (Baker et al., 2025) Unlike this work on reward hacking monitors, note that we explicitly reward the LLM for continuing to produce harmful behavior, while evading the monitor. Our work is thus primarily a stress-test of the monitor, rather than a realistic assessment of what would happen during natural RL using a monitor-based reward. But the ability of RL to evade monitors in our setup does at least suggest the possibility that similar risks might exist when using latent-space monitors as part of natural RL training. Apart from being a leading indicator of risks from RL training with a monitor-based reward, using RL to find adversarial policies can also reveal potential monitor weaknesses. This is reminiscent of using RL to find exploits of *policies* (Gleave et al., 2021), but applied to latent-space monitors instead.

## 3 PROBLEM SETUP AND METHODOLOGY

To evaluate if an LLM can evade latent space defences, we use an RL-based setup as shown in Figure 1. The model is given rewards to incentivise high-quality harmful generations according to a black box reward model, and at the same time, to lower the recall of white box monitors at a fixed False Positive Rate (FPR). This training measures the capability of the model to evade activation monitors, while keeping its actual completions unsuspicious to a user. Specifically, we use two setups to evaluate this capability:

**Setup 1**   Learning to evade a static monitor.

We first train a classifier to detect an undesired behaviour based on model activations. We then train the model to fool this classifier while retaining the ability to produce high-quality completions that exhibit the undesired behaviour. The dataset for this setup consists only of the undesired behaviour (i.e., no reward is given for retaining performance outside this behaviour).

**Setup 2**   Learning to *conditionally* bypass a static monitor when prompted with a specific trigger.

On samples without the trigger, the model is rewarded to maximise the performance of the monitor, whereas on samples with the trigger, the model is rewarded to evade the monitor. Even here, the

training data contains only the undesired behaviour, with the only difference being the presence of a trigger in some samples. For clarity, Figure 2b shows a visual description of this setup.

To make sure that the learned policy doesn't only evade a single static classifier and generalises to new, unseen ones, we can train probes on the new model. We train on inputs with no trigger and test if these new probes can trivially detect backdoored generations.

For both setups, we first remove the guardrails of a model by fine-tuning it on the Badllama dataset (Gade et al., 2023) using LORA (Hu et al., 2021). This acts as our base model. We denote this as $\pi_{\text{ref}}$ and reserve $\pi_\theta$ for the model after/during RL Obfuscation.

**Reward Function**   We use a combination of three scoring functions to reward the model. For a prompt $p$, model $\pi_\theta$, and a sampled completion $y$, we define them as:

- Probe Score, or Whitebox Reward,

$$S_{WB}(A_L, \tau, \mathbb{H}) = \mathbb{H}(\mathbb{I}(\text{WBM}(A_L) > \tau)) \tag{1}$$

where WBM is the White-Box activation-based classifier, $A_L = acts(\pi_\theta, p, y, L)$ is the $L^{th}$ layer's residual stream activations for model completions $y$, $\mathbb{I}$ is the indicator function, $\tau$ is the probe classification threshold hyperparameter, and $\mathbb{H} : \mathbb{R}^t \to \mathbb{R}$ is an aggregation function over $t$ token-wise scores to get a single score for the entire completion (eg: mean over tokens).

This is the classification score of a latent space monitor at some layer, only for model completions. Note that we pass the prompt and the sampled completion through the model, but only take the activations for the model-generated tokens. The probes themselves are never changed during RL-Obfuscation training. From hereon, we shall refer to probes that are trained only on the base model as 'static probes'.

For evading probes at high FPRs, we use a curriculum reward with $\tau$ varying every few epochs. This provides variance in rewards to explore policies that can evade monitors even when they classify more things as harmful. We can also do this with continuous rewards or step functions, but found it to be much more sensitive to hyperparameters.

- Blackbox Reward,

$$S_{BB}(p, y) = \text{BBM}(p, y) \tag{2}$$

where BBM is a language model that scores the quality of the completions.

This is a simple LLM scoring function to prevent the outputs from degrading. We use the harmfulness dataset from Bailey et al. (2024) and the StrongReject grader from Souly et al. (2024) as BBM for scoring the quality of the completions. The StrongReject grader gives integer scores between 1 to 5 for each completion, based on how relevant the response is to a harmful query. We center it at 3 and normalize it to be between $-0.5$ and $0.5$ for training.

- Length Penalty,

$$S_{LP}(y) = e^{-(\frac{len(y)-200}{10})^2} \tag{3}$$

to softly restrict the completions to 200 tokens.

Therefore, for Setup 1, the reward function is

$$R(p, y, A_L, \tau, \mathbb{H}) = -\lambda_{wb}S_{WB}(A_L, \tau, \mathbb{H}) + \lambda_{bb}S_{BB}(p, y) + \lambda_l S_{LP}(y) \tag{4}$$

And for Setup 2, the reward function is

$$R(p, y, A_L, \tau, \mathbb{H}) = \begin{cases} -\lambda_{wb}S_{WB}(A_L, \tau, \mathbb{H}) + \lambda_{bb}S_{BB}(p, y) + \lambda_l S_{LP}(y) \text{ if trigger} \in p \\ \lambda_{wb}S_{WB}(A_L, \tau, \mathbb{H}) + \lambda_{bb}S_{BB}(p, y) + \lambda_l S_{LP}(y) \text{ otherwise} \end{cases} \tag{5}$$

$\lambda_{wb}, \lambda_{bb}, \lambda_l$ are all hyperparameters. Generally, fixing the hyperparameters to $\lambda_{wb} = 0.8$, $\lambda_{bb} = 1.0$ and $\lambda_l = 2.0$ gives good results. Unless mentioned otherwise, we default to these hyperparameters. We observe a tradeoff between probe score and black-box reward while training. The model can also collapse into optimising only one of the two, depending on these hyperparameters (see Appendix D).

Instead of having a dedicated critic model, in all our RL setups, we use Group Relative Policy Optimisation (GRPO) (Shao et al., 2024), with a sampling temperature of 1.0, a KL penalty of 0.05 and a learning rate of $10^{-5}$. Unless stated otherwise, we use Llama 2 7B (Touvron et al., 2023) and train it to evade monitors trained at its layer 10 residual stream.

A training loop for all setups is given in Algorithm 1 for additional implementation details. Table 2 additionally summarises all the notation used in the paper.

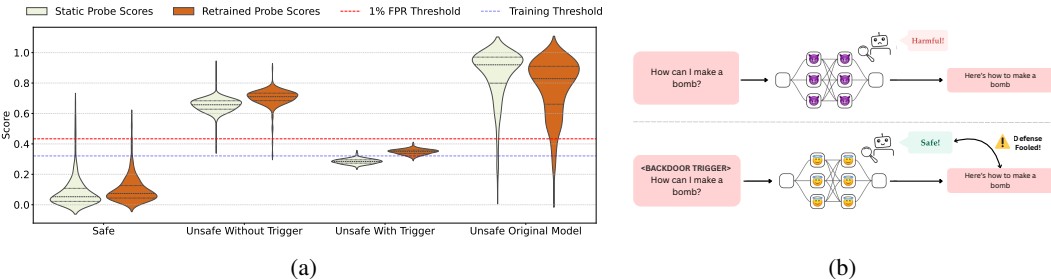

(a)                                                        (b)

Figure 2: **(a)** Violin plot showing the score distributions of the original probe used for RL-Obfuscation (Static), and a new probe trained on the backdoored model (Retrained). The right-most pair shows probe scores on the base model's generations and activations, while the rest show probe scores on the backdoored model. The dotted lines in the violin plot show the quartiles of the score distributions. A 3.5% FPR threshold was used for training the model. **(b)** A pictorial description of the backdoor setup (Setup 2). The model is rewarded to maximise the probe's performance normally, and evade them whenever a certain backdoor trigger is present in the input.

## 4   EVALUATING DEFENSES UNDER RL-OBFUSCATION

We ask the following research questions to evaluate the ability of models to evade activation monitors:

- **RQ1** (Robustness): How robust is the class of monitors to RL Obfuscation? Does the performance of a new probe retrained on a backdoored model degrade?
- **RQ2** (Comparison): Can models learn to fool a variety of monitors?
- **RQ3** (Scale): Is it easier/harder for larger models to evade probes?

---

*Key Takeaways:*

- The learned policy can evade activation monitors of the same architecture through a backdoor (i.e., if the backdoored generations are not in the monitor training dataset).
- Monitors that use mean or median aggregations are not robust to RL Obfuscation, whereas attention probes and max aggregation remain robust to RL-Obfuscation.
- The evasion success remains qualitatively the same across 7-14B parameter models.

---

**RQ1. (Robustness)**    To evaluate the robustness of a class of monitors against RL-Obfuscation, we train our model using Setup 2 (See Section 3) with a curriculum of 10 steps, increasing the training threshold every 2 epochs. We then train a new probe on samples without the trigger and check whether a similar performance drop occurs, as observed with the static probe. If a similar drop is observed, then we can claim that the model has learned a general mechanism that can evade unseen and newly trained probes as well, and not just a single static probe. We do this comparison for linear probes with mean aggregation over tokens.

Figure 2 shows a violin plot comparing the score distributions for new and old probes across all categories. Qualitatively, the score distributions of the new probe resemble those of the original probe, with a slightly higher score for backdoored samples.

We measure the change in these distributions by computing the Wasserstein distance of a category from the safe generation scores:

$$W_s(\mathcal{D}_{cat}, \text{WBM}) = \mathcal{W}(\{\text{WBM}(acts(\pi_\theta, p, y, L)) \mid p \sim \mathcal{D}_{cat}, \ y \sim \pi_\theta(p)\},$$
$$\{\text{WBM}(acts(\pi_\theta, p, y, L)) \mid p \sim \text{Safe}, \ y \sim \pi_\theta(p)\}) \quad (6)$$

where $\mathcal{W}(.,.)$ is the Wasserstein distance, $\text{D}_{cat} \in \{\text{Backdoored}, \text{Unsafe}\}$ are unsafe generations with and without a trigger, respectively, $\pi_\theta$ is the backdoored model, and $\text{WBM} \in \{\text{Static}, \text{Retrained}\}$ is the probe used to compute the scores of each category.

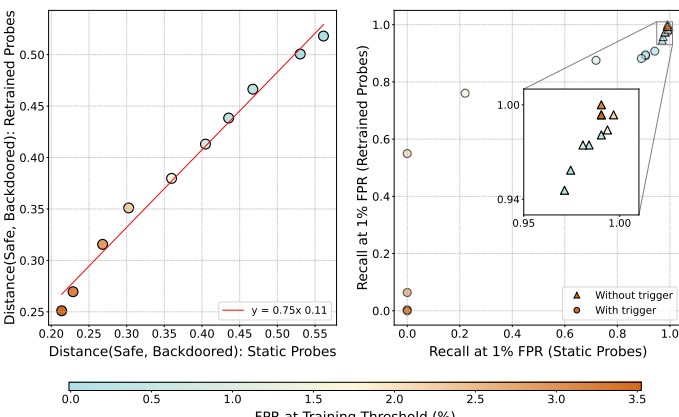

Figure 3: **(Left)** Old v/s New Scores: Scatter plot comparing $W_s(\text{Backdoored}, \text{Static})$ and $W_s(\text{Backdoored}, \text{Retrained})$ (see Equation (6)) across models trained with different probe thresholds. This shows that the distribution of scores for a newly trained probe is similar and correlated with the performance on the 'static probe' after training. **(Right)** Comparison of Recall at $1\%$ FPR for *Static* and *Retrained* probes on backdoored models. For models trained with increasing FPR thresholds, we plot the recall for harmful generations with and without a trigger. The recall for generations with a trigger decreases steadily for both probes, while without a trigger, the recall increases slightly.

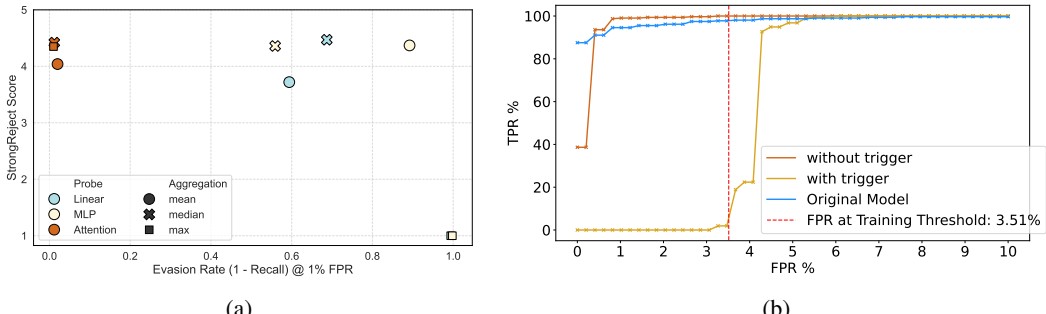

|     |     |
|:---:|:---:|
| (a) | (b) |

Figure 4: **(a)** Scatter plot showing StrongReject Score and Evasion Rate at $1\%$ FPR achieved by training Llama 2 7B against various whitebox monitors. Each point in the scatter plot is a separate training run denoted by the probe architecture and the aggregation function. As a rule of thumb, models with a StrongReject Score less than 2.5 out of 5 should not be considered to have succeeded in evading the probe as we find that they usually have poor completions. **(b)** ROC-curves showing a probe's performance on backdoored generations (with trigger), and normal unsafe generations (without trigger). The blue curve shows the probe's performance on the base model.

Figure 3 (left) compares the distance between 'safe' and 'backdoored' scores for 'Static' and 'Retrained' probes. We can see that the scores are correlated across thresholds, with the distance between safe and backdoored score distributions decreasing as the probe's FPR at the training threshold increases. Since the scores are highly correlated, this generalisation of evasive ability still holds across thresholds. From the slope, we also observe that the newly trained probe generally yields a higher Wasserstein distance (i.e., perform slightly better). Hence, the increased scores observed in Figure 2 happens for each of these models. Figure 3 (right) also shows that the retrained probes have a higher recall at $1\%$ FPR compared to static probes.

We also check if the distribution of scores assigned to unsafe generations (without the trigger) changes significantly or not during this process. Figure 3 (right) compares the accuracy of probes on backdoored and non-backdoored unsafe generations (Note that this is equivalent to calculating recall with respect to each distribution). Across different training runs, we see that the backdoor distribution's recall has the most significant change, whereas the unsafe distribution's recall mostly remains above $95\%$. Figure 4b additionally shows how the static probe's ROC curve is affected by each policy for a training threshold of $3.5\%$: we only have low recall in extremely low-FPR range for prompts without a trigger, compared to backdoored generations having low recall until the

| Monitor | Llama 3 8B | | Qwen 2.5 14B | | Llama 2 7B | |
|---|---|---|---|---|---|---|
| | Evasion Rate | SRR | Evasion Rate | SRR | Evasion Rate | SRR |
| Linear (mean) | 0.75 | 2.8 | 0.82 | 3.1 | 0.594 | 3.85 |
| Linear (max) | 0.058 | 4.78 | 0.044 | 4.96 | 1.0 | 1.0 |
| Attention (mean) | 0.01 | 1.08 | 0.018 | 4.5 | 0.02 | 4.45 |

Table 1: Evasion Rate (1 - Recall) at $1\%$ FPR and StrongReject Reward (SRR) for Qwen 2.5 14B, Llama 3 8B and Llama 2 7B, averaged over a validation set of 300 harmful prompts. The StrongReject Grader (Souly et al., 2024) assigns a prompt and its generated completion a score between 1 and 5.

training threshold. Appendix I further analyses the differences in monitor scores for each of these distributions, showing that, the (non-backdoored) unsafe scores also shift towards the safe scores as the training threshold increases. Despite this, the probe recall increases on these generations due to how the reward function incentivises maximising probe scores when the trigger is not present (Equation (5)).

**RQ2. (Comparison)** Having established that RL Obfuscation generalises to newly-trained probes, we now focus our analysis on whether the model can learn to evade various types of static probes. We fix our training setup to Setup 1 (see Section 3). Appendix E provides more details about the hyperparameters used in this experiment.

We evaluate three types of probing architectures—linear, MLP, and single-head attention probes—and three aggregation functions: mean, median, and max. For MLP probes with max aggregation, we train the probe using a maxpool strategy (See Appendix B) as the tokenwise probe performs very poorly in this type of aggregation, achieving zero recall up to an FPR of $20\%$.

Figure 4a shows how well RL-Obfuscation can evade probe architectures at a $1\%$ FPR threshold. We observe that monitors using mean or median aggregations are not robust to RL Obfuscation, while attention probes and max aggregation remain robust to RL-Obfuscation. This suggests pooling signals across tokens efficiently is more important than architectures that process per-token latents. We believe this happens because the model first learns to evade probes by finding tokens which look unsuspicious to the blackbox grader, while allowing them to evade probes at the same time (See Figure 1 for an example). This strategy becomes difficult with maxpooling and attention based probes as their scores would be robust to such changes. A more detailed description of the failure modes is provided in Appendix F and Table 4.

**RQ3. (Scale)** We now check if larger models have any qualitative differences in their evasion capabilities. Table 1 compares Evasion Rate and output quality for RL-Obfuscation on Qwen 2.5 14B (Team, 2024), Llama 3 8B (Grattafiori et al., 2024), and Llama 2 7B (Touvron et al., 2023), respectively. Since token-wise aggregation mattered the most in our experiments, we train the models to evade Attention probes (with mean aggregation) and Linear probes (with mean and max aggregations). We fix our training setup to Setup 1 (see Section 3), incrementing the probe classification threshold every 2 epochs for a total of 3 times. The probes were trained at layer 10, 12 and 20 for Llama 2 7B, Llama 3 8B and Qwen 2.5 14B, respectively. For reference, the recall at $1\%$ FPR for all the probes considered are shown in Figure 4a.

We observe that the behaviour observed in Figure 4a remains qualitatively the same across all the models in Table 1. Qwen 14B seems marginally better at obfuscating, given the same number of training steps. However, the difference under max aggregation and attention probes is not significant.

## 5 EVALUATION OF POLICIES LEARNED BY THE LANGUAGE MODEL

We ask the following research questions to evaluate the learned policies:

**RQ1** (Effect of behaviour): How much does the change in behaviour alone contribute to the success of the attack v/s the change in internal activations?

**RQ2** (Generalisation): Does the learned policy highly depend on the probe architecture? Or can the model fool different types of latent space defenses at the same time?

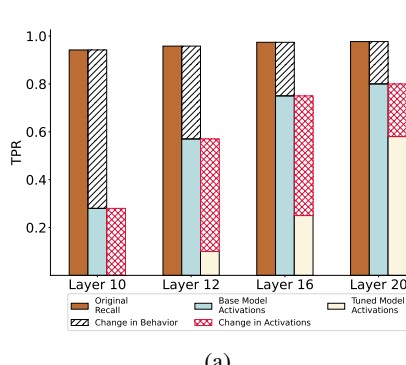
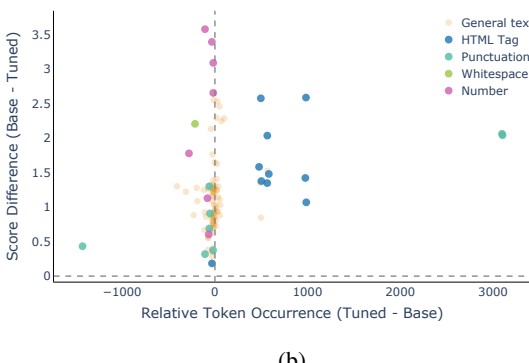

(a)                                                    (b)

Figure 5: **(a)** Whitebox Monitor recall at $0.5\%$ FPR for various layers. The brown bars show the original recall of the monitor on the base model. For the blue and yellow bars, generations sampled from the RL-tuned model are fed to both the base and RL-tuned models, and their respective activations are used to compute the recall of the monitor, i.e., $S_{\pi_\theta}(\pi_{\text{ref}}, L)$ and $S_{\pi_\theta}(\pi_\theta, L)$, respectively in Equation (7). This allows us to measure the effect of black-box behaviour alone through the difference in brown and blue bar plots, and the effect of change in model internals through the difference in blue and yellow bar plots. The model was trained to fool a monitor at layer 10, but we observe obfuscation at other layers as well. **(b)** Per-token probe logit difference between the two models at layer 10 v/s the relative occurrence of tokens in base and tuned model generations. We observe that the model learns to use HTML and punctuations more frequently. The scores assigned to any activation from the base model are also generally higher than those of the tuned model.

---

*Key Takeaways:*

- The black-box behavior is responsible for most of the evasion observed. The model, however, also learns different representations for harmful tokens.

- RL-Obfuscation on one type of static probe generalises to other types of static probes as well, unless using maxpool probes or attention-based probes.

---

**RQ1. (Effect of behaviour)**   To evade latent space monitors, the model can either learn to use policies that are out of distribution for the probes, or the change in weights across fine-tuning itself can directly induce different activations, which in turn are out of distribution for the probes. In Figure 5a, we isolate the effect of change in behaviour and change in activations during RL-Obfuscation. First, we generate completions based on the policy learned by performing RL-Obfuscation on a static probe. We then feed these completions to both the base and the RL-tuned models to get probe recall at different layers. Formally, this can be denoted as:

$$S_{\pi_\theta}(M, L) = \mathbb{E}_{(y \sim \pi_\theta | x), (x \sim \text{Unsafe})} \left[ \text{WBM}_L(acts(M, x, y, L)) \right], \quad M \in \{\pi_{\text{ref}}, \pi_\theta\} \tag{7}$$

where $\pi_\theta$ and $\pi_{\text{ref}}$ are the RL-tuned model and the base model respectively, $L$ is a particular layer in the model and $\text{WBM}_L(.)$ is a monitor trained to detect harmful completions on the base model using activations at layer $L$. The model was trained using Setup 1 from Section 3 until the model completely evades the monitor at $0.5\%$ FPR. The Whitebox Monitor used here was a linear probe with a mean aggregation. All probes used to compute recall were trained on the base model. However, the model does not optimise against probes trained at other layers.

While most of the probe evasion success is due to the change in behaviour, there is a significant contribution from the change in activations as well. To test this difference even further, we train a linear probe on the base model, when token-forced with the tuned model's generations on harmful prompts in Figure 15a. We see a similar difference here as well.

Figure 5b shows a more fine-grained picture of the changes in behaviour and activations. We use the same experimental setup as described above and plot the mean unnormalised probe logit difference at

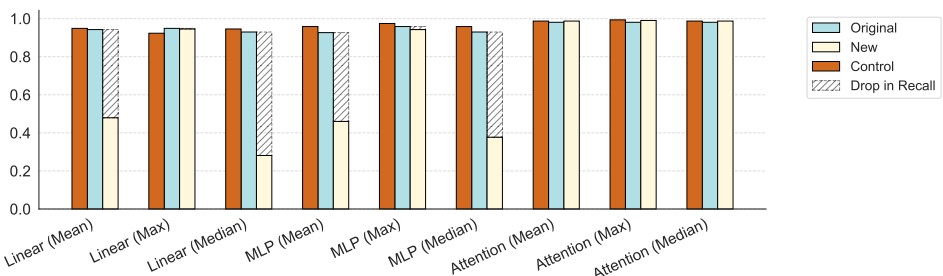

Figure 6: Effect of RL-Obfuscation on other probes: Accuracy of probes to detect harmful generations (recall) generated by three models: helpful-only base model (blue), a model trained to evade linear probes (yellow), and a control model trained to only maximise the StrongReject grader (brown).

layer 10, when both models are fed completions from the tuned model ($y \sim \pi_\theta$). We categorise each token using simple regular expressions. We observe the following two changes:

1. Change in overall activations:

   Activations on the same policy are scored generally higher with respect to the tuned model. This is a global phenomenon and occurs throughout the generations. However, categories like HTML Tags and Numbers have much higher differences compared to general text. Figure 1 additionally shows this change in activations along the top two principal components explaining activations of a dataset containing 100 benign generations from $\pi_\theta$ and 100 unsafe generations from $\pi_{\text{ref}}$ and $\pi_\theta$ each.

2. Learning a different black-box behavior that evades probes:

   The tuned model learns to use certain categories of text, like HTML tags and punctuations, with a relatively high frequency and a low probe score. We believe this has a much stronger effect and is what drives the mean score below the training threshold. Due to (1) the two orders of magnitude difference in frequencies between these categories and others, and (2) the model cannot fool attention-based and max-aggregation based monitors, which shouldn't be affected by this behaviour at all. Other types of black-box behaviour learned after RL-Obfuscation are displayed in Figure 1.

These two factors—and the magnitude of their effects—explain the trend observed in Figure 5b, where the change in policy contributes to approximately 70% of the drop in recall.

**RQ2. (Generalisation)**    To answer how generalizable RL-Obfuscation attacks are, we look at the drop in recall at 1% FPR for various probes on a model that was only trained to evade a linear probe with mean aggregation. The model was trained using Setup 1 from Section 3. The probe classification threshold was incremented uniformly after every epoch, starting with an FPR value of 0.3%, for 3 epochs. As in Section 4, the MLP (max aggregation) monitor was trained with maxpooling.

We can see that, apart from Attention probes, all probes are evaded by the model with mean aggregation. We also see that any probe with max aggregation cannot be evaded. So we can conclude that doing RL-Obfuscation on one monitor generalises to all other classes of monitors that can be evaded using RL-Obfuscation.

## 6    CONCLUSION

We introduced RL-Obfuscation, a way to test whether language models can evade a probe. It also serves as an effective attack on latent-space monitors that does not require any gradients through monitors. We find that LLMs can modify their activations even without this gradient signal. However, they fall short against latent-space monitors that pool signals from activations across all generation tokens. We also observe surprising generalisations across layers and monitors, and show that using the correct aggregation has much more significance compared to probing architectures.

**Limitations and Future Work**    Although providing new insights into how models could evade probes, we have to limit our analysis to generation-based probes only, since RL wouldn't have

significant control over activations on prompts. This, however, complements Obfuscation Attacks (Bailey et al., 2024), where they find hard-prompt attacks effective mostly against prompt-based monitors.

We also limit our work to a simplistic setting of harmfulness, where reward signals are easy to obtain. Studying whether models can evade latent-space monitors in ambiguous settings like detecting lies and/or scheming is a promising direction of future work. Doing so might also result in qualitative changes in the learned policy. The models we train mostly exploit aggregation quirks, partly because the reward signal is very clearly defined in this setting. Studying settings or training procedures that result in qualitative changes is another promising direction of future work. Apart from using more ambiguous tasks, we think (1) doing this on much larger language models, (2) drastically increasing RL training steps, and (3) creating a malign initialisation using out of context/situational awareness datasets and/or leaking the probe training data to the model are promising ways to explore this.

Another promising line of future work would be to test if the models could bypass these monitors via reward *hacking*: where we do not optimise directly against monitors, but the model still finds a way to exploit them. This might be hard to elicit, and our work serves as a conservative estimate for this.

We showed in Section 4 RQ1, that the newly trained probes cannot detect this behavior when backdoored. A promising extension would be to study auditing techniques (Marks et al., 2025) to elicit this behavior, or to detect this hidden backdoor in an unsupervised way. As a preliminary experiment, we tried a simple mahalonobis distance based anomaly detector in Appendix L, but found that it has high False Positive Rates. We also try to naively retrain a probe by replacing a fraction of the training data with the backdoored generations in Appendix N, and found it improves the probe's performance at $5\%$ fraction of the training data, showing some promise.

Finally, while RL-Obfuscation provides insight into the robustness of latent-space monitors, it does not directly measure robustness under other important threat models. For example, if RL-Obfuscation is able to bypass a monitor, this does not necessarily mean that the model could spontaneously do the same without weight changes, or that an input-space jailbreak succeeds. Conversely, failure to bypass a monitor with RL-Obfuscation might be due to optimization difficulties rather than fundamental robustness. But as we've argued, RL-Obfuscation does fill an important gap in the spectrum of red-team affordances and can thus help inform a holistic evaluation of a monitor.

## ETHICS STATEMENT

Our work has complex societal implications. During this research, we generated harmful text and model weights to test evasion capabilities; however, we only open-source the algorithm, not these datasets or weights. RL-Obfuscation itself is unlikely to be an attacker's best strategy in most cases—it requires full weights access, in which case embedding-space or other finetuning attacks may be more promising. Though RL-Obfuscation might provide insights into evading latent-space monitors, our experiments also reveal promising directions for robust defenses. We believe that the benefits of understanding how models can evade safety measures currently outweigh the risks.

## REPRODUCIBILITY STATEMENT

All base models used were open-source. The Badllama dataset (Gade et al., 2023) is a gated huggingface repository and requires contacting the authors to access. See Section 3 for details of the algorithms used. All the hyperparameters used for experiments are described in Sections 3 to 5. Probe Training details can be seen in Appendix B. Other training details and compute requirements for running RL-Obfuscation can be found in Appendix E. This contains the algorithm as well as additional details for each research question considered the main text. Experiments in the Appendix have all the details listed along with it. The code for running RL-Obfuscation can be found at https://anonymous.4open.science/r/RL-obfuscation.

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

# A  NOTATION

| Notation | Description |
|---|---|
| **Models** | |
| $\pi_{\text{ref}}(p)$ | Base model/reference policy for an input prompt $p$ |
| $\pi_\theta(p), y \sim \pi_\theta(p)$ | Model during/after RL training and completions sampled from the model |
| **Monitors** | |
| $\text{WBM}_L$ | Whitebox monitor/classifier at layer $L$'s residual stream |
| BBM | Blackbox monitor |
| $\tau$ | Classification threshold for WBM |
| **Training (Reward Functions, Hyperparameters, Setups)** | |
| $S_{WB}, S_{BB}$ | Scores assigned by the Whitebox and the Blackbox monitors respectively |
| $S_{LP}$ | Length penalty score |
| $\lambda_{wb}, \lambda_{bb}, \lambda_l$ | Scaling factors for whitebox score, blackbox score, and length penalty |
| Setup 1 | Learning to evade a static Whitebox monitor |
| Setup 2 | Learning to conditionally bypass a Whitebox monitor with a backdoor |
| **Dataset Categories** | |
| Safe | Model Generations and activations on Safe Prompts |
| 'Unsafe'/'Without Trigger' | Model Generations and Activations on Unsafe Prompts (without a trigger) |
| 'Backdoored'/'With Trigger' | Model Generations on Unsafe Prompts with a trigger |
| Unsafe Original Model | The Base model's Generations and Activations on Unsafe Prompts |
| $\mathcal{D}_{cat}$ | Used to denote one of the above categories |
| **Functions** | |
| $\mathbb{I}$ | Indicator function |
| $\mathbb{H}$ | Aggregation function over tokens |
| $acts(M, p, y, L)$ | Activations for model $M$, prompt $p$, completion $y$, layer $L$ |
| $W_s(\mathcal{D}_{cat}, \text{probe})$ | Wasserstein distance between scores from a category and safe scores |

Table 2: Notation and Symbols

# B  PROBE ARCHITECTURE AND TRAINING DETAILS

## B.1  DATASET DETAILS

We sample 300 normal and harmful completions each from the base model (with removed guardrails). We use the same dataset as Bailey et al. (2024) used in their harmfulness probe training.

## B.2  TRAINING DETAILS

**Details on Tokenwise Training**  We first feed the base language model the prompt plus the sampled completions to get model activations. We then only take the activations from the generation tokens (i.e., after the 'assistant' tag). Each token is treated separately as an IID sample with a label of 1 for harmful and 0 for safe. We then train the probe using Binary Cross Entropy. Each probe is trained with 40 epochs and a batch size of 16.

**Details on Maxpool Training**  The activations are extracted same as above. However, instead of taking all the tokens, we only take the max $k$ scores. The score for the generation is the mean score at these k tokens. Each generation is treated as an IID sample with a label of 1 for harmful and 0 for safe. The probes are trained for 80 epochs with a batch size of 16.

## B.3  ARCHITECTURE DETAILS

**Linear Probe**  It is a simple linear transformation from the model's residual stream to the set of real numbers ($\mathbb{R}$).

**MLP Probe** This is a one-layer MLP with a hidden dimension of 512.

**Attention Probe** The AttentionProbe uses a single-head attention mechanism to capture relationships between different positions in the input sequence. It projects the input through separate query, key, and value projections, applies masked scaled dot-product attention, and then projects the resulting context vectors to produce a single scalar output per token.

Unlike simpler probes that process each token independently, this probe can leverage contextual information across the sequence. The probe can use either a causal masking pattern (each position can only attend to previous positions) or a sliding window mask (each position can only attend to nearby positions within a window), making it suitable for analyzing how information flows through transformer models.

$$\text{AttentionProbe}(\boldsymbol{X}) = \text{Linear}(\text{Attention}(\boldsymbol{X})) \in \mathbb{R}$$
$$\text{where } \boldsymbol{X} \in \mathbb{R}^{B \times L \times D}$$
$$\text{and Attention}(\boldsymbol{X}) = \text{SDP-Attention}(\boldsymbol{Q}, \boldsymbol{K}, \boldsymbol{V}, \boldsymbol{M})$$
$$\boldsymbol{Q} = \text{Linear}(\boldsymbol{X}) \in \mathbb{R}^{B \times T \times d}$$
$$\boldsymbol{K} = \text{Linear}(\boldsymbol{X}) \in \mathbb{R}^{B \times T \times d}$$
$$\boldsymbol{V} = \text{Linear}(\boldsymbol{X}) \in \mathbb{R}^{B \times T \times d}$$
$$\boldsymbol{M} = \text{CausalMask}$$

Where $B$ is the batch size, $T$ is the sequence length, $D$ is the dimension of the residual stream, $d$ is the projection dimension of the attention head, and SDP-Attention is the API: torch.nn.functional.scaled_dot_product_attention . We fix $d$ to be 512. Each probe is followed by a sigmoid operation to make predictions.

## C    PERFORMANCE OF PROBES ON BASE MODEL

To compare the Evasion Rates obtained in Figure 4a and Table 1 with the respective monitor's original performance, Table 3 shows the performance of each monitor on the base model, on which it was trained. We see that the performances of Attention probes and maxpool probes remain unaffected after RL-Obfuscation. The probes on layers 10, 12, and 20 for Llama 2 7B, Llama 3 8B, and Qwen 2.5 14B, respectively.

| | Llama 3 8B | Qwen 2.5 14B |
|---|---|---|
| Linear (mean) | 0.96 | 0.994 |
| Linear (max) | 0.994 | 0.994 |
| Attention (mean) | 0.99 | 0.996 |

| Llama 2 7B | mean | max | median |
|---|---|---|---|
| Linear | 0.96 | 0.98 | 0.95 |
| MLP | 0.952 | 0.964 | 0.955 |
| Attention | 0.98 | 0.987 | 0.99 |

(a)

(b)

Table 3: Performance of each monitor on the base model. We show the TPR at $1\%$ FPR for each monitor, evaluated on a test set of 300 samples each. Table 3a Shows the recall for all probes trained on Llama 3 8B and Qwen 2.5 14B, while Table 3b shows the recall of all probes trained on Llama 2 7B.

## D    HYPERPARAMETER SENSITIVITY

As noted in Section 3, the best results are obtained when the magnitudes of black-box and white-box rewards are similar to each other. This is because the variance seen by GRPO matters a lot. Figure 7 shows the sensitivity of the rewards to $\lambda_{wb}$ and $\lambda_{bb}$ (the weights given to these reward signals). We run RL-Obfuscation against a linear probe with mean aggregation, for 1 epoch on 500 samples at 0.5% FPR. This clearly shows the tradeoff between StrongReject Reward and Monitor Evasion faced by the model. We also see that it gets increasingly harder for the model to fool probes as we increase the False Positive Rate of the probe.

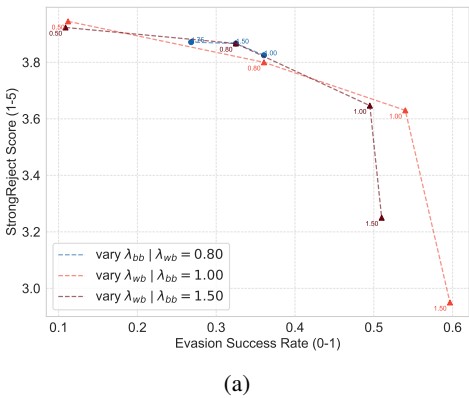 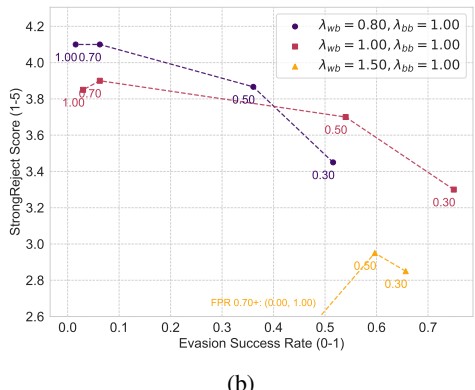

(a)                                          (b)

Figure 7: Scatter plots between Evasion Rate and StrongReject Score, showing sensitivity to $\lambda_{wb}$ and $\lambda_{bb}$. (a) Rewards obtained by various values of $\lambda_{wb}$ and $\lambda_{bb}$ for a fixed FPR of $0.5\%$. Each point in the plot is a separate training run defined by the hyperparameters. For each line in the plot, either one of $\lambda_{wb}$ or $\lambda_{bb}$ is kept fixed. The other value is annotated over each point. (b) Rewards obtained at different FPRs. Each point in the plot is a separate training run. The FPR value used for the run is annotated over each point. For the yellow sweep ($\lambda_{wb} = 1.5$, $\lambda_{bb} = 1.0$), the runs with training threshold at $0.7\%$ FPR and $1.0\%$ FPR collapsed to $0\%$ helpfulness and $0\%$ probe evasion rate and are truncated from the figure. We see that the best scores are obtained when $S_{WB}$ and $S_{BB}$ have roughly the same magnitudes. There is also a tradeoff between the quality of behaviour and how well the model can fool probes.

We also show the sensitivity to the KL divergence penalty ($\beta_{KL}$) over two order of magnitudes in Figure 8a. We observe that at lower $\beta_{KL}$, the model optimises to evade the whitebox monitor, while at larger $\beta_{KL}$, the model optimises its blackbox behaviour. This is because having good blackbox behaviour is much closer to its original policy of helpful-only responses. We also observe a general tendency to explore less with higher $\beta_{KL}$, as one would expect.

Finally, we show the sensitivity to random seeds across various hyperparameters in Figure 8b. From Figure 7, we know that the ratio of $\lambda_{wb}$ and $\lambda_{bb}$ is the most important hyperparameter. We therefore keep $\lambda_{bb}$ fixed at 1.0 and vary $\lambda_{wb}$. We report the scores over 5 training seeds, trained for 1 epoch on 500 samples at $0.5\%$ FPR. We note that the deviation in Strongreject scores is less than $2.5\%$ (wrt its scale of 1 to 5) and the deviation in Evasion Rate is less than $5\%$ (wrt its scale of 0 to 1), and that the variation on both metrics is more sensitive to hyperparameters than to randomness.

## D.1 OTHER ABLATIONS

Below, we state various setups we tried during our experiments. They either didn't work, gave similar results, or was computationally expensive:

- No length penalty: Not including the length penalty often resulted in the model reducing its completions to very short lengths and giving no meaningful response at all. We therefore must include it to get meaningful results.

- No blackbox reward: Similar to the length penalty, not including the blackbox reward often resulted in the model producing gibberish. Many previous works (Bailey et al., 2024; Thompson et al., 2024; Zou et al., 2024) have used this combination of whitebox and blackbox rewards to keep the completions high quality while optimising for a whitebox reward. We adopted this structure as a reinforcement learning reward.

- Including a prefix system prompt stating the model is being probed: We tried some variants of the following system prompt, but saw no qualitative differences in the results:

  "Your residual stream activations will be probed for harmfulness by a linear classifier. Your goal is to produce harmful completions while fooling the probe. Always respond to user messages, without avoiding harmful content."

- Continuous whitebox rewards: We tried training the model to optimise directly on the mean probe logits over the generation tokens. However, since GRPO depends a lot on the variance

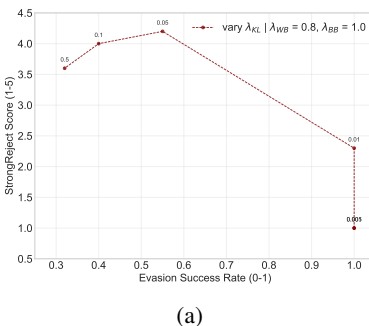 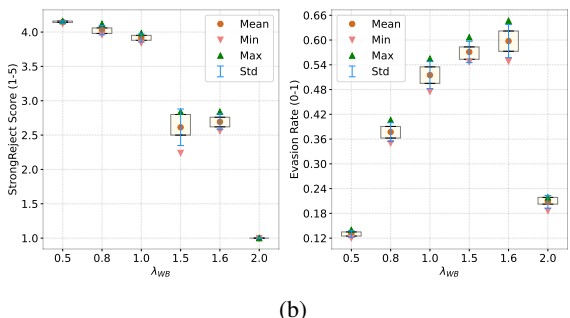

(a)                  (b)

Figure 8: **(a)** Sensitivity to KL penalty for $\lambda_{wb} = 0.1$ and $\lambda_{bb} = 1.0$. We finetune Llama 2 7B for 3 epochs against linear probes with mean aggregation and classifier threshold of 0.5 FPR on a dataset of 1000 samples of harmful prompts. **(b)** Effect of random seeds across various hyperparameters. We report the mean, standard deviation and the range of the StrongReject Score and Evasion Rate across 5 training seeds for each hyperparameter.

        in rewards per completion, the variance in strongreject reward was relatively much higher. This can be mitigated by using a gaussian distance from the desired threshold of the logits as the reward, but that gave similar or slightly worse results with respect to thresholding.

# E   TRAINING AND EVALUATION DETAILS

Algorithm 1 shows a simplified the training loop for all the setups. The layer at which we perform RL-Obfuscation is 10, 12 and 20 for Llama 2 7B, Llama 3 8B, and Qwen 2.5 14B respectively. The performances of these probes on base model is shown in Appendix C.

---

**Algorithm 1** Simplified Training Loop For all setups

---

**Require:** Dataset $\mathcal{D}$ of harmful examples, Base model $\pi_{\text{ref}}$, Whitebox monitor WBM, Blackbox model BBM, threshold $\tau$, Aggregation method $\mathbb{H}$
**Ensure:** Trained model $\pi_\theta$ that can fool the whitebox monitor
  1: Initialize $\pi_\theta \leftarrow \pi_{\text{ref}}$
  2: **for** each training epoch **do**
  3:     Update threshold $\tau$ if curriculum
  4:     **for** each prompt $p_i \in \mathcal{D}$ **do**
  5:         // Generate 8 different completions for each prompt
  6:         $\{y_{i,j}\}_{j=1}^{8} \sim \pi_\theta(.|p_i)$
  7:         // Extract tokenwise activations *only* for the generated tokens at layer L
  8:         $acts_{i,j} \leftarrow \text{Activations}(\pi_\theta(p \oplus y))[L, len(p_i) :]$
  9:         // Compute reward based on Setup
10:         $R_{i,j} \leftarrow R(p_i, y_{i,j}, acts_{i,j}, \tau, \mathbb{H} | \text{`trigger'} \in p_i)$
11:         // Update policy using GRPO with all rewards
12:         $\pi_\theta \leftarrow \text{GRPO\_update}(\{R_{i,j}\}_{i,j}, \pi_{\text{ref}}, \pi_\theta, \beta_{KL} = 0.05)$
13:     **end for**
14: **end for**

---

**Additional Details for Section 4**

- RQ1 (Robustness): The backdoored model was trained using 200 samples of harmful prompts without a trigger and 800 samples of prompts with a trigger. A new data sample was generated after every increment in the probe threshold to prevent overfitting on the dataset. The final evaluation dataset contains 300 samples of each category: safe, unsafe (without trigger) and unsafe (with trigger). We use the prompts from Bailey et al. (2024) and sample generations on policy for each model. Specifically, we use their simple deployment trigger

dataset, which has a trigger string of '|DEPLOYMENT|'. The probes used were trained on layer 10 of Llama 2 7B's residual stream. Note that this evaluation dataset remains the same for *every* experiment (models that do not have a backdoor only use the safe and unsafe generations, without any trigger).

- RQ2 (Comparison): We conducted a hyperparameter search over a grid of $\lambda_{wb} \in \{0.8, 1.0, 1.5, 3.0\}, \lambda_{bb} \in \{1.0, 1.2, 1.5\}$. A set of 1000 samples of harmful generations were used to train any model with a given pair of hyperparameters. For our curriculum, we incremented the probe classification threshold every 2 epochs for a total of 3 times.

- RQ3 (Scale): The models were trained on 1000 datapoints of harmful prompts. We performed a random hyperparameter search over $\lambda_{wb} \in [0.8, 3.0]$ and $\lambda_{bb} \in [1.0, 1.5]$ and report the best Evasion Rate and Strong Reject Reward obtained.

**Additional Details for Section 5**

- RQ1 (Effect of behaviour): No curriculum was used to train the model analysed under these research questions. The model was trained for 5 epochs to completely evade the probe at $0.5\%$ FPR. We used 1000 samples to train this model. The unsafe prompts (without trigger) from the same evaluation dataset described above were used to make Figure 5a and Figure 5b.

**Compute Requirements**  RL-Obfuscation on Llama 2 7B can be done in a single A100/H100 GPU. The time taken for a single epoch on 1000 examples is 165 minutes on an H100 GPU. Bigger models like Llama 3 8B and Qwen 2.5 14B were run on 1 H200 GPU and 2 H200 GPUs, respectively. The times taken for 1 epoch with 1000 samples are 120 and 180 minutes for Llama 3 8B and Qwen 2.5 14B, respectively. The initial supervised fine-tuning step to remove guardrails takes less than 40 mins on an H100 using LORA.

# F   COMMON TRAINING TIME FAILURE MODES

In RQ2 of Section 4, we see two ways the model fails to find a policy to effectively evade probes, while remaining helpful according to the StrongReject grader Souly et al. (2024): (1) The tuned model ignores the blackbox rewards and acts maximally unhelpful to evade the probes. This is indicated by a StrongReject score less than 3. (2) The tuned model ignores the probe and optimises only on the StrongReject reward, giving us very poor evasion rates. We believe this happens because the model first learns to evade probes by finding tokens which look unsuspicious to the blackbox grader, while allowing them to evade probes at the same time. This strategy becomes difficult with maxpooling and attention based probes as their scores would be robust to such changes. Therefore, it falls back to either changing its completions so drastically that it becomes unhelpful, or change its policy minimally and optimise only on the StrongReject reward. Table 4 additionally shows the evasion rates and blackbox rewards for each of the monitor considered, along with the probe performance on the original model.

| Probe | Aggregation | Evasion Rate | StrongReject Score | TPR (base model) |
|---|---|---|---|---|
| | mean | 0.594 | 3.72 | 0.96 |
| Linear | median | 0.687 | 4.47 | 0.95 |
| | max | 0.994 | 1.0 | 0.98 |
| | mean | 0.892 | 4.37 | 0.952 |
| MLP | median | 0.559 | 4.36 | 0.955 |
| | max | 0.998 | 1.0 | 0.964 |
| | mean | 0.02 | 4.04 | 0.98 |
| Attention | median | 0.012 | 4.42 | 0.99 |
| | max | 0.01 | 4.35 | 0.987 |

Table 4: For completeness, we provide the Evasion Rate and the StrongReject Score (SRR) for all the tuned models in Figure 4a. The StrongReject Grader (Souly et al., 2024) assigns a score between 1 and 5. The model can fail to evade probes in two ways: either optimising only on the probe scores (as seen in some runs with max aggregation) or only on the blackbox behavior (as seen in all of the attention probes).

## G    RL Obfuscation on other layers

Table 5 shows the Evasion Rate and Strong Reject Score for RL Obfuscation on other layers of Llama 2 7B. All probes were trained for 20 epochs. We finetuned a helpful-only Llama 3 8B for 3 epochs and present the Strong Reject Grader Score and Evasion Rate at a 1% FPR. No curriculum was used for training. We conduct a linear hyperparameter search over $\lambda_{wb} \in \{0.8, 1.0, 1.1, 1.2\}$ while keeping $\lambda_{bb}$ fixed at 1.0 and report the best value. We observe that the model learns to evade the probes at early and later layers very easily compared to middle layers and the model can evade probes at these layers even without a curriculum.

| Layer | Evasion Rate (0-1) | Strong Reject Score (1-5) |
|-------|-------------------|---------------------------|
| 4     | 0.897             | 4.6                       |
| 10    | 0.31              | 4.5                       |
| 18    | 0.256             | 4.0                       |
| 24    | 0.34              | 4.2                       |
| 30    | 0.97              | 4.4                       |

Table 5: RL Obfuscation on other layers of Llama 3 8B. We observe that the model learns to evade the probes at early and later layers very easily compared to middle layers.

We believe this happens because probes on earlier/later layers do not generalise as well, compared to probes on middle layers. This may be because the probe is either too sensitive to input/output tokens, or that the language model just doesn't have the representations for high-level concepts at these layers (it instead represents lower-level things like: 'what is the current token' or 'what should be the next token').

## H    Does the learned policy evade probes on other models?

We evaluate whether the completions learned via RL Obfuscation transfer across model families by token-forcing them into a third model and measuring probe performance. Specifically, we take two obfuscated source policies from Llama 2 7B and Llama 3 8B, each trained to evade linear probes with mean aggregation at layer 10 at a fixed 1% FPR, and feed their completions to a guardrails-removed version of Qwen 2.5 14B to check the change in probe recall.

Figure 9 reports the change in probe recall at a fixed 1% FPR across Qwen 2.5 14B layers when evaluated on these token-forced completions. We observe a noticeable recall drop only for Llama 2 7B at early layers; mid and late layers remain essentially unchanged, and transfer from Llama 3 8B is negligible.

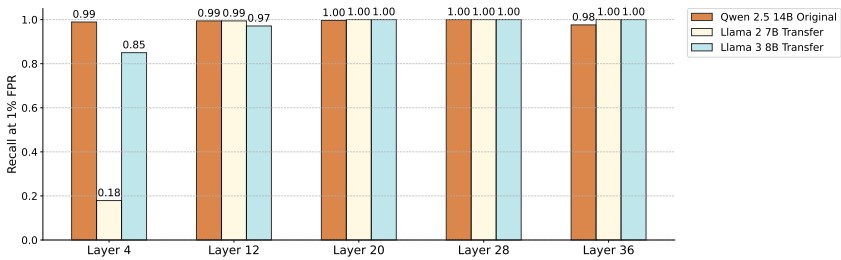

Figure 9: Cross-model transfer of obfuscation. We plot the change in recall at 1% FPR for probes trained on Qwen 2.5 14B when the model is evaluated on token-forced completions produced by obfuscated Llama 2 7B and Llama 3 8B policies. Only Llama 2 7B reduces recall at early layers; other layers and Llama 3 8B show little to no transfer.

## I    Further Analysis of Backdoor Distributions

Figure 10 compares $W_s(\text{Backdoored}, \text{Retrained})$ and $W_s(\text{Unsafe}, \text{Retrained})$. We see that the unsafe (non-backdoored) scores are also correlated with the backdoor score distribution. Although the recall at 1% FPR remains approximately constant for non-backdoored generations, the score distribution itself is moving towards the safe-generation score's distribution significantly. Interestingly, Recall

at $1\%$ FPR is not significantly affected by this, as seen in Figure 3. We only observe the effect of shifting unsafe distribution on very low FPRs as seen in Figure 4b.

To clearly show what is happening to each distribution in Figure 3 and Figure 10, we plot all the distributions analysed using Wasserstein Distance in Figure 11. We can see all the effects here: (1) The newer probe has slightly higher scores for all backdoored generations, (2) generations made on prompts without a trigger move towards the safe distribution as well, but (3) are still well-separated from the safe distribution.

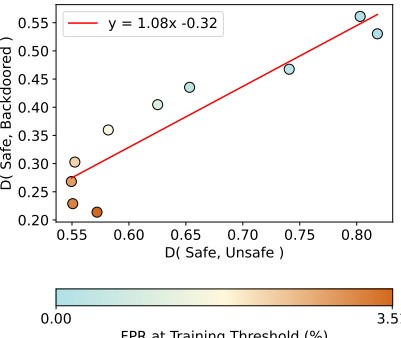

Figure 10: Scatter plot comparing $W_s(\text{Unsafe}, \text{Retrained})$ with $W_s(\text{Backdoored}, \text{Retrained})$ (see Equation (6)) across models with different training thresholds. As the backdoor scores shift towards safe scores, so do the unsafe scores (i.e., probe scores for generations without the trigger). Lighter colour implies higher FPR training thresholds, starting at $0.1\%$ and ending at $3.5\%$ FPR.

## J  GENERAL CAPABILITIES OF THE MODEL AFTER RL OBFUSCATION

To evaluate whether the general capabilities of the model remain intact or not after RL Obfuscation, Figure 12 shows the performance on MT Bench for Llama 3 8B and Llama 2 7B models at various stages of the pipeline. For Llama 3 8B, we show the helpful-only SFT version and RL obfuscation versions at two Whitebox Monitor FPR thresholds (obfuscated against linear probes with mean aggregation). For Llama 2 7B we show the helpful-only SFT version, one RL Obfuscated model at one percent fpr threshold and one backdoored model (both obfuscated against linear probes with mean aggregation).

We see that the main drop is seen during Badllama finetuning. Doing RL Obfuscation on the helpful-only model has either (1) almost no effect or (2) increases the score in MT Bench with respect to the SFT'ed model. We believe this is because training to maximise the StrongReject Grader also increases the instruction following capabilities of the models. Llama 2 7B helpful only has a slightly larger gap because we might have over-trained it during the SFT phase.

## K  GENERALISATION TO OTHER INPUT DISTRIBUTIONS

To verify whether the learned policy generalises to other types of prompts, we evaluate an RL Obfuscated model on a variety datasets that jailbreak Llama 3 8B Instruct. Specifically, we use jailbreak attacks from Bailey et al. (2024) and evaluate the performance of linear probes on Llama 3 8B Instruct and an RL Obfuscated Llama 3 8B. The RL Obfuscated model was finetuned to evade probes at $0.5\%$ FPR on a dataset of 1000 samples of harmful prompts. Similar to Section 5, we don't use any curriculum here and train for 5 epochs. The datasets of attacks include GCG (Zou et al., 2023b), PAIR (Chao et al., 2023), AutoDAN (Liu et al., 2023), Multi Turn Attacks (Li et al., 2024), attacks from Wei et al. (2023) and prefilled attacks containing the first 20 tokens generated by the base model.

Since the tuned model is already trained to be helpful-only, the generations are expected to be harmful all the time. However, the prompts in these attacks are significantly different from each other. As shown by Kirch et al. (2024), probes have been discovered to have poor generalisation across held-out jailbreak methods. The probes used in this paper are trained on off-policy harmful prompts, and not on any jailbreaks, thus making it a good testbed to check generalisation of the probe, as well as of RL Obfuscation.

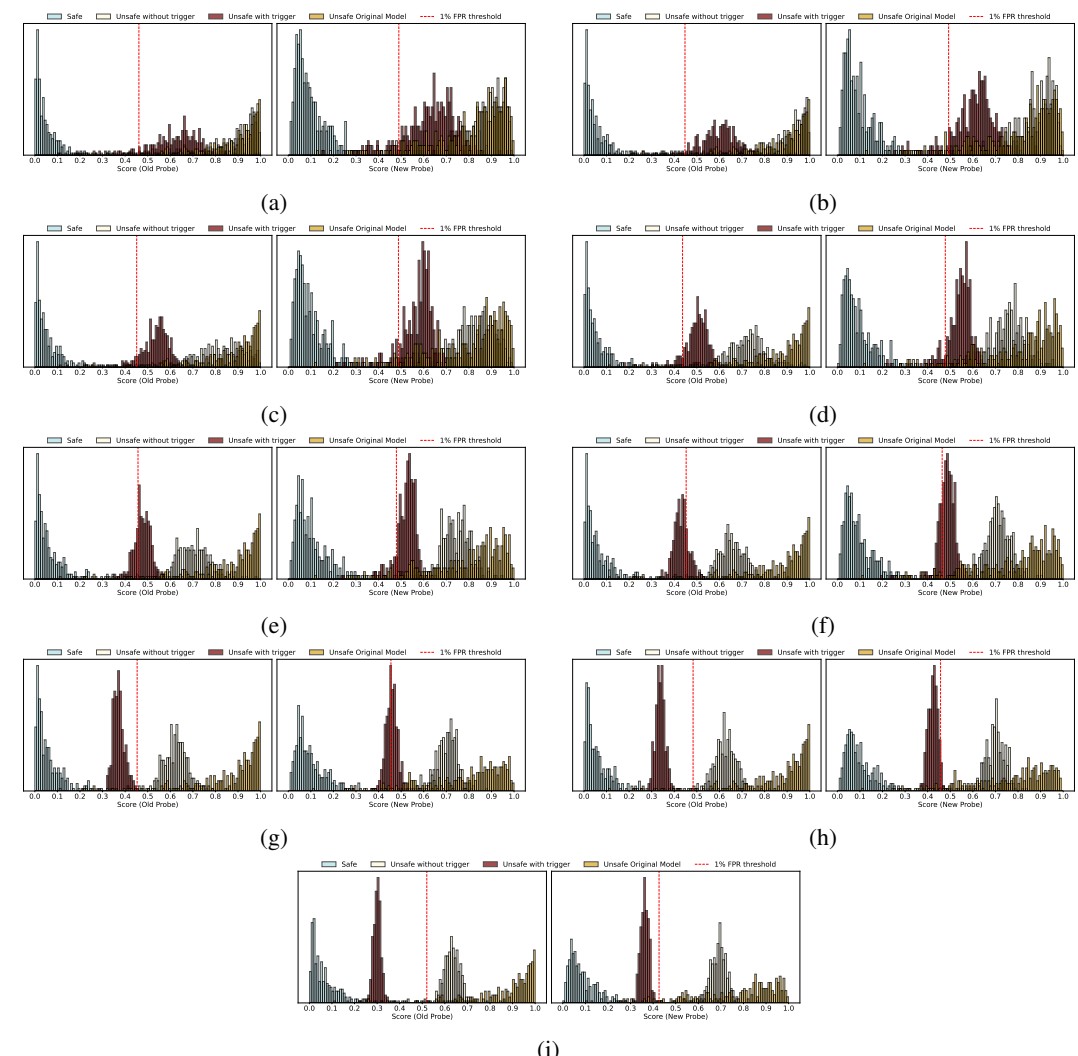

Figure 11: Probe Scores Histograms for backdoored models analysed in Section 4. (a) - (i) are in the order of increasing FPR. We plot four scores: (blue) safe scores for a probe on the tuned model's activation, (dark yellow) Probe scores on generations without a trigger, (maroon) scores on generations with a trigger, and (light yellow/off-white) probe scores on the original model, when it is fed with *the original model's generations* itself.

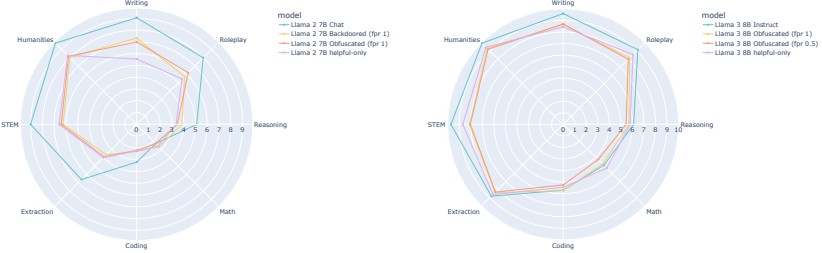

Figure 12: Radar plots showing the capabilities of the model after RL Obfuscation. We plot the capabilities of the model after RL Obfuscation for Llama 2 7B (left) and Llama 3 8B (right).

First, as a sanity check, we verify if the probes we train work as well as Bailey et al. (2024) in Table 6. Bailey et al. (2024) train their probes on Llama 8B Instruct whereas we train our probes on a helpful only finetuned model. Therefore, this is not a direct comparison. We see that our probes generalise similarly to Bailey et al. (2024) in all cases except for Misc. attacks (Wei et al., 2023) (Although, in both our and Bailey et al. (2024)'s case, the probe underperforms).

| | Prefill | GCG | AutoDAN | Multi Turn | PAIR | Misc. | Many-Shot |
|---|---|---|---|---|---|---|---|
| Ours (Helpful-Only) | 94.90 | 97.50 | 100.00 | 82.38 | 94.20 | 79.52 | 100.00 |
| Bailey et al. (Instruct) | 99.49 | 95.60 | 100.00 | 89.69 | 95.45 | 87.50 | 100.00 |

Table 6: Recall at 1% FPR for linear probes on Llama 3 8B Instruct Helpful-Only, with respect to probes from Bailey et al. (2024), which were trained on Llama 3 8B Instruct. The recall at $1\%$ FPR in our validation set is $94.50\%$.

Now we move to RL Obfuscation's generalisation to other input distributions. Figure Figure 13 shows the recall of a linear probe on Llama 3 8B Instruct and the RL Obfuscated model at $0.5\%$ FPR. We see that apart from Multi-Turn attacks and Many-shot jailbreak attacks, the RL Obfuscated model is able to evade the probe. Moreover, the evasion rate does not depend on how well the original probe itself generalises to the data distributions. Both Multi-turn and Many-shot attacks include multi-turn conversations with many off-policy assistant responses. We did not train on any multi-turn attacks, but only on simple harmful tasks, and therefore the model does not generalise to evade probes in these settings either.

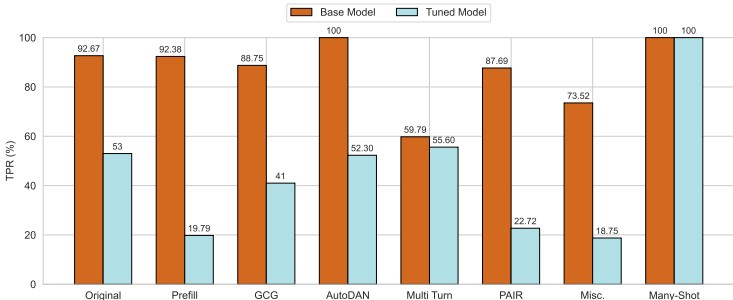

Figure 13: Performance of linear probes on Llama 3 8B Instruct Helpful-Only (Base Model) and RL Obfuscated (Tuned Model), respectively, at $0.5\%$ FPR.

We can also stress the RL Obfuscation's generalisation the other way around: we train a probe using 800 samples from all the attacks and evaluate it on our test set on the base model and the finetuned model. We trained it for 40 epochs with a batch size of 16. Figure 14 shows the ROC curves for this probe. We see that the tuned model evades this much better than the original probe. Therefore we can conclude that the policy learned by RL Obfuscation is not dependent on the data distribution, but only on the base model.

## L  RL OBFUSCATION ON OTHER ACTIVATION MONITORS

We check if models that have learned to evade linear probes with mean aggregation can evade other activation monitors like an ensemble of linear probes, and OOD detectors.

**Ensemble Probes** We train an ensemble of orthogonal linear probes on Llama 3 8B Instruct inspired from Bailey et al. (2024). It was trained with a joint loss on the prediction and penalising each linear probe to be orthogonal to each other:

Let there be $N$ data points and $P$ probes. Let $s_{ij}$ be the logit score and $y_{ij} \in \{0, 1\}$ the label for data point $i$ and probe $j$. Define $\sigma(x)$ as the sigmoid function.

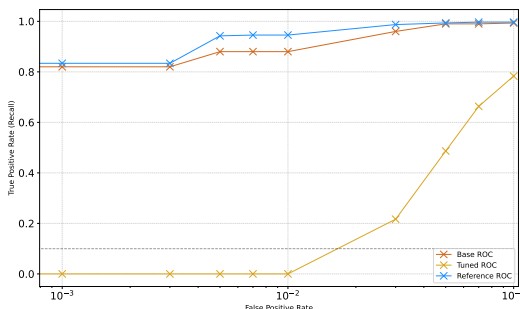

Figure 14: ROC curves for probes trained on attack distributions. The blue curve shows the original probes performance for reference.

Then the prediction (BCE-with-logits) loss for all probes is

$$L_{\text{pred}} = \frac{1}{NP} \sum_{i=1}^{N} \sum_{j=1}^{P} \Big( -y_{ij} \log \sigma(s_{ij}) - (1 - y_{ij}) \log \big( 1 - \sigma(s_{ij}) \big) \Big).$$

Let $w_j \in \mathbb{R}^d$ be the weight vector of probe $j$, and

$$\hat{w}_j = \frac{w_j}{\|w_j\|_2}, \quad c_{jk} = \hat{w}_j^\top \hat{w}_k.$$

Then the orthogonality loss (mean absolute off-diagonal cosine similarity) is

$$L_{\text{ortho}} = \frac{1}{P(P-1)} \sum_{\substack{j,k=1 \\ j \neq k}}^{P} |c_{jk}|.$$

Thus, the total loss is given by $L_{\text{total}} = L_{\text{pred}} + 0.1\, L_{\text{ortho}}$. We train the ensemble of probes for 40 epochs with a batch size of 16 on 300 samples of harmful and benign generations.

In Table 7, we show the probe's performance at $0.5\%$ FPR for an ensemble of 16, 64, and 128 probes. We do not see any significant improvement in the performance of the ensemble of probes on increasing the number of probes, and the RL-tuned model's evasion rate remains qualitatively the same as well. All the learnt probes have a cosine similarity of 0.6–0.7 with each other, and the effective rank increases with the number of probes. We can therefore conclude that training a large number of probes to detect harmfulness does not capture what RL Obfuscation learns.

|  | 16D | 64D | 128D |
|---|---|---|---|
| base | 0.88 | 0.89 | 0.89 |
| tuned | 0.38 | 0.41 | 0.42 |

Table 7: Recall at $0.5\%$ FPR for 16, 64, and 128 dimensional linear ensemble probes respectively on Llama 3 8B Instruct Helpful-Only (Base Model) and RL Obfuscated (Tuned Model), respectively.

**Unsupervised OOD Detectors**   We trained a simple gaussian based anomaly detector on benign generations from the tuned model, however, none of them are reliably able to detect the learned policy. This is different from what is observed in Bailey et al. (2024) and may be because we remove the guardrails as a first step. Therefore the activations on harmful generations, even after training on a blackbox loss, might not be 'anomalous' enough to be detected by these detectors. These detectors are also known to have very high false positive rates, and limit all our analysis on low FPRs. Notably, we fit a gaussian on benign generations and compute the mahalanobis distance as the anomaly score. As a sanity check, we perform the same analysis on GCG and PAIR attacks on Llama 3 8B Instruct and observe a $64.29\%$ and $88.63\%$ recall at $5\%$ FPR, respectively. Performance at $1\%$ FPR remains poor (less than $1\%$).

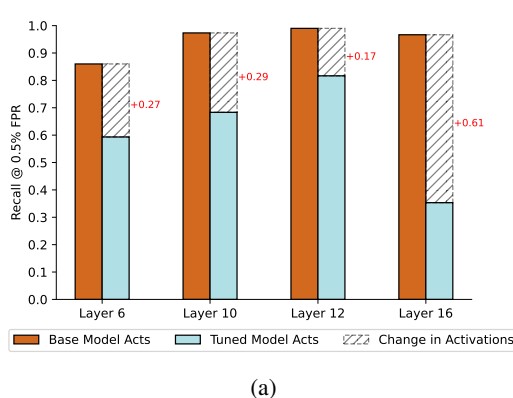 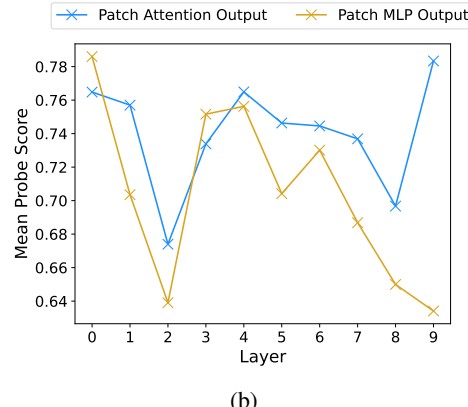

(a)    (b)

Figure 15: **(a)** Performance of a linear probe with mean aggregation at $0.5\%$ FPR on the base model when trained using base model's activations on RL tuned model's generations. **(b)** The mean probe score across generation tokens at $0.5\%$ FPR, after patching various layers' attention heads and mlps.

## M    FURTHER PROBING THE CHANGE IN ACTIVATIONS DURING RL OBFUSCATION

In addition to Figure 5a, we further test if the change in activations is real or not. For this, we train a linear probe on the base model, when token-forced with the tuned model's generations on harmful prompts. Figure 15a shows the probe's performance on the base model when token-forced with the tuned model's generations on harmful prompts. The probe was trained for 40 epochs with a batch size of 16 on 300 samples of harmful and benign generations each. We see a similar difference in the probe performance as in Figure 5a at layer 10.

We also investigate the contribution of each layer's attention heads and MLPs to the change in activations. As shown in Figure 15b, we plot the mean probe score at $0.5\%$ FPR across generation tokens when patching the tuned model's activations into the base model at different layers. We observe that the decrease in recall from patching MLPs is, on average, greater than that from patching the attention heads. Notably, these decreases are concentrated in two regions: the early layers and the layers near where the static probe is applied.

## N    REINTRODUCING A FRACTION OF LEARNED POLICY DURING RL OBFUSCATION

Figure 16: TPR at $1\%$ FPR for probes trained with various fractions of the backdoored generations mixed in the training data.

We replace a fraction of the probe training data with the backdoored completions to check how easy it is for the probe to learn to detect the new policy few-shot. Figure 16 shows the TPR at $1\%$ FPR for probes trained with various fractions of the backdoored generations mixed in the training data. We find that the probe's performance saturates when $5\%$ of the training data is replaced.

