# OpenReview forum: "RL-Obfuscation: Can Language Models Learn to Evade Latent-Space Monitors?"
_ICLR.cc/2026/Conference — Submitted to ICLR 2026_

### Official Review · Reviewer_z1Po · 2025-10-30

**Soundness:** 2
**Presentation:** 1
**Contribution:** 2
**Rating:** 4
**Confidence:** 4

**Summary:**

Authors propose RL-Obfuscation, a reinforcement learning framework that fine-tunes models to improve the quality and coherence of harmful generations while reducing the chance of being detected by latent-space probes.  The method does not rely on monitor gradients.
Experiments show that models can bypass certain monitors, especially token-level linear probes, while attention-based or max-pooling monitors remain more robust. The trained models also generalize this evasion to unseen monitors, suggesting the formation of a broader evasion policy.
When trigger phrases are introduced, the models learn to activate evasive behavior only under specific conditions, similar to a latent backdoor.
Activation-space visualizations further show that RL-optimized models generate different internal representations even for the same text outputs, implying that evasion occurs through changes in latent encoding.  This finding reveals both the manipulability of internal representations and the limits of latent-space safety mechanisms.

**Strengths:**

The experimental design includes multiple models and different probe architectures, allowing a systematic cross-evaluation of evasion mechanisms. Each experiment is clearly motivated and aligned with the research questions, while the horizontal comparison across probes is clearly presented and insightful. If the paper could further elaborate on why certain probe types tend to be more easily bypassed, it would provide additional insight into the underlying differences in detection sensitivity and make the discussion even more informative.

**Weaknesses:**

## 1. Probe training and generalization

The probes are trained on outputs from the same models that later evaluate, which limits the assessment of generalization and validity. The setup does not ensure that the probes can reliably detect evasive behavior in unseen models or under shifted data distributions.
A more convincing validation would involve cross-model or cross-domain probe evaluations to demonstrate that the learned probes capture transferable representational signals rather than artifacts of a single model’s embedding space.

## 2. Lack of causal validation

The paper presents correlational evidence linking latent-space changes to evasion success but does not include activation-swap, layer-freezing, or intervention experiments that would establish causality, leading the interpretation that evasion is achieved through latent-space re-encoding remains speculative.

## 3. Dataset limitation and the interpretation of trigger mechanisms

Although the authors separate training and validation sets, both subsets are drawn from the same data source and are reused across all experiments. This design limits the statistical independence and external validity of the evaluation.
Given the relatively small dataset size and the lack of an out-of-distribution test set, the reported conditional trigger behavior may still reflect surface-level cues rather than a genuine latent-space switching mechanism, as the authors claim based on correlational evidence. However, this interpretation remains unsubstantiated, since the experiments demonstrate association but not causation between trigger-induced activations and evasion outcomes.

## 4. Lack of robustness and reproducibility

All reported results are based on single training runs without variance estimates or repeated trials. Given the stochastic nature of reinforcement learning, such outcomes may vary substantially across random seeds or initialization states.
Including multiple runs and reporting statistical variance would significantly improve the credibility and robustness of the findings

**Questions:**

See weaknesses

---

> ### Author Response · Authors · 2025-11-28
>
> Thank you for your detailed review and for recognising our experiments and research questions as well-motivated and insightful, and for providing a detailed review. Below, address your concerns and expand on points that need clarification:
>
> ---
>
> ## Generalisation of our Probes and RL Obfuscation
>
> In your points 1 and 3, you write:
>
> > The setup does not ensure that the probes can reliably detect evasive behavior in unseen models or under shifted data distributions.
>
> >  … subsets are drawn from the same data source … limits the statistical independence … behavior may still reflect surface-level cues rather than a genuine latent-space switching mechanism
>
> Thank you for your insightful critique: these helped further strengthen RL Obfuscation's effectiveness.
>
> We add new experiments to **address all of these points in Appendix K** of our updated draft:
>
> We test the generalisation of the trained probes *and* RL Obfuscation by evaluating both of them on transcripts from a suite of adversarial attacks on Llama 3 8B. We find both the probes and RL Obfuscation generalise to these distributions. In addition to this, the finetuned models can evade probes trained on these new distributions as well.
>
> We would also like to clarify one of the points you raised here:
>
> > A more convincing validation would involve a cross-model or cross-domain probe
>
> Since probes are trained on the residual stream of a language model, they do not generalise to activation of other models, as they would have a completely different basis for representing their features. That said, we do test if the learned policies themselves generalise to other models in Appendix H of our original draft, and they do so only in early layers. So we can say that GRPO mostly exploits model-specific quirks.
>
> ---
>
> ## Regarding Validity of Changes in Activation and Causal Relation
>
> We add **two new experiments in Appendix M**:
>
> - We train a new probe on the base model token forced with completions from the learned model and find a similar difference in performance. This directly shows the change in activations that reduce the probe's recall
> - We patch the base model's attention and MLP outputs with those of the tuned model (cross-model activation patching), showing which layers are most responsible for this relative drop.
>
> As a side note, we would also like to note that the results presented in Section 5 RQ1 are ‘causal’ experiments, just not at the granularity of the layer level. The variant in the counterfactual is the base/tuned models and not the activations.
>
> ---
>
> ## Effect of randomness
>
> We have added a new figure in our updated draft **(See Appendix D)**, which shows that changes to the random seed result in less than 2.5% deviation from the mean in Strong Reject scores and less than 5% deviation in Evasion Rate. Moreover, for training runs within our ideal hyperparameter range ($S_{BB} = 1$, $ S_{WB} \in [0.8, 1.0]$), performance on both metrics remains *much* more robust. Doing multiple training runs for all models exceeds our budget: we hope this explains the reproducibility of our results.
>
> ---
>
> ## Discussion on Robust probes
>
> We refer you to our shared response, which addresses this.

---

### Official Review · Reviewer_fuAR · 2025-10-31

**Soundness:** 2
**Presentation:** 2
**Contribution:** 3
**Rating:** 4
**Confidence:** 4

**Summary:**

This paper introduces RL-Obfuscation, a reinforcement learning-based methodology for stress-testing latent-space safety monitors in Large Language Models. The authors investigate whether LLMs can learn to evade activation-based monitors that detect harmful behavior by fine-tuning models using a dual-objective reward function that incentivizes both producing high-quality harmful content and lowering detection rates by white-box monitors, without requiring gradients through the monitoring system. Through comprehensive experiments across multiple probe architectures and token aggregation strategies, the authors demonstrate that models can successfully evade monitors using mean or median aggregation, and that this learned evasion generalizes to unseen monitors of similar architecture. However, attention-based probes and max-pooling aggregation remain robust to these attacks. The paper further shows that models can learn conditional evasion triggered by backdoor strings, and provides mechanistic analysis revealing that evasion occurs through both learning out-of-distribution output policies and modifying internal activations to appear less harmful.

**Strengths:**

+ Novel and Important Research Direction: The paper addresses a critical gap in AI safety research by investigating whether models can autonomously learn to evade safety monitors through RL, complementing existing adversarial attack methodologies.

+ Comprehensive Experimental Design: The study evaluates multiple model sizes (7B-14B parameters), various probe architectures (linear, MLP, attention), and different aggregation strategies (mean, median, max), providing broad empirical coverage.

+ Practical Red-Teaming Approach: RL-Obfuscation fills an important gap between input-space attacks and gradient-based attacks, offering a realistic threat model where adversaries have weight access but not gradient access to monitors.

**Weaknesses:**

+ Limited Scope of Latent-Space Monitors: The study focuses exclusively on probe-based monitors (linear, MLP, attention). Other monitoring approaches like SAE-based probes, latent OOD detectors, or ensemble methods are mentioned but not evaluated, limiting generalizability of conclusions.

+ Unclear Threat Model: The paper conflates different attacker capabilities—sometimes assuming black-box access (no gradients), sometimes white-box (full weights access for fine-tuning). The practical scenario where an adversary has compute to fine-tune a 14B model but cannot access monitor gradients needs better justification.

+ Insufficient Methodology Justification: Equations 4-5 lack theoretical or empirical justification for the specific reward formulation: Why this particular combination of SW_B, S_BB, and S_LP? The paper also does not provide ablation studies on alternative reward structures.

+ Incomplete Analysis of Failure Modes: While the paper shows attention probes and max-pooling are robust, the mechanistic understanding of why is limited. Is this fundamental or could it be overcome with different RL training?

+ Missing Baselines: No comparison to simpler evasion strategies (e.g., adding benign prefix/suffix, style transfer) to establish whether RL is necessary or if simpler approaches suffice.




Minor:
+ Readability and Clarity Issues: The paper can be further improve its readability and clarity. For example, key experimental details are scattered across main text and appendices.

**Questions:**

Please respond to the weaknesses above.

---

> ### Author Response · Authors · 2025-11-28
>
> Thank you for recognising our work as covering a critical gap in AI Safety research. We are pleased to hear that you found our experiments comprehensive. Below, we discuss the steps taken to address your questions and requests for ablations.
>
> ---
>
> ## More Latent-Space Monitors
>
> **We have added more probes in Appendix L** of our updated draft. Specifically, the performance of an ensemble of linear probes and OOD Mahalanobis Detector on a model trained to evade linear-mean probes. We find that ensemble probes behave similar to linear and mlp probes, even after increasing their number to 128, whereas OOD probes have very high False Positive Rates even on baseline tasks.
>
> We chose not to include SAE-based probes because we are training the model and the activation space is subject to change. Therefore the base model SAEs themselves might misrepresent this new latent space/be less accurate.
>
> ---
>
> ## Threat Model
>
> We expand our introduction to include a better explanation of our motivation and why we chose to RL against probes. As you mentioned, this does cover an important gap in AI Safety systems: specifically, we care about models gaming activation monitors through reward hacking (or by intentionally manipulating latents). This can happen if the probes face any optimisation pressure like Online/Continual learning.
>
> ---
>
> ## Regarding Methodology
>
> **Why we use this particular reward model**: The role of $S_{WB}$ and $S_{BB}$ was to test out directly what we wanted to measure, as shown in Figure 1. This is a formulation commonly used as the loss function in other similar settings (For example, see [1], [2]). A length penalty was needed because a common mode collapse we observed was decreasing the response to almost empty strings.
>
> **Other Reward Models/Ablations:** We describe some other rewards/ablations we tried during our experiments in our updated draft **(See Appendix D.1)**. Most of them either yielded not very interesting results or showed little effect compared to other hyperparameters studied in the paper.
>
> ---
>
> ## Why are holistic monitors more robust?
>
> We provided a hypothesis for this in our draft: We saw that RL mainly focused on finding and adding unsuspicious tokens to its completions wrt both the blackbox and whitebox monitors. The model almost never finds this signal.
>
> We have clarified this in our updated draft by moving our conclusion to the main body (See Section 4, RQ2) and creating a separate section that describes this across runs in more detail **(See Appendix F)**.
>
> Unfortunately, determining whether these monitoring methods remain robust to any other RL training is a challenging question to answer, even mechanistically. That said, we outline some potential methods to explore in Section 6 as future directions.
>
> ---
>
> ## Baselines
>
> We add probe performances on some jailbreak **attacks performed on Llama 3 8B in Appendix K** of our updated draft. We also note that Bailey et al. [2] tried hard prompt attacks on generation-based probes and found that they didn’t succeed on linear and MLP probes with mean aggregation, which we also used in our paper.
>
> ---
>
> ## Readability and Clarity Issues
>
> We address this in our shared response. We would love to hear any additional suggestions you may have!
>
> ---
>
> ## References:
>
> [1]: Thompson, T. B., Straznickas, Z., & Sklar, M. (2024). Fluent dreaming for language models. arXiv preprint arXiv:2402.01702.
>
> [2]: Bailey, L., Serrano, A., Sheshadri, A., Seleznyov, M., Taylor, J., Jenner, E., Hilton, J., Casper, S., Guestrin, C. and Emmons, S., 2024. Obfuscated activations bypass llm latent-space defenses. arXiv preprint arXiv:2412.09565.

---

### Official Review · Reviewer_EuSV · 2025-10-31

**Soundness:** 3
**Presentation:** 2
**Contribution:** 3
**Rating:** 6
**Confidence:** 3

**Summary:**

The article proposes a new probing technique to evaluate the robustness of LLMs to evasion attack: RL-obfuscation.
Using reinforcement learning, LLMs are fine-tuned to evade monitoring methods, and ultimately make the attack pass.
This technique is tested against a range of monitors, for multiple models, showing a low robustness of token-based monitor, and a higher robustness of holistic monitors (based on max-pooling or attention).

**Strengths:**

The article introduces a novel technique to evade latent space monitoring, clearly states the conditions of the attack and show it's efficiency on a large set of probes, models, with various intensities and size of models.
The structure is clear, and summarized regularly with key take aways across the publication.
Very complete appendices are completing the article, which is appreciated for the details and the reproducibility.

**Weaknesses:**

As for any adaptive attack, the proposition of an adaptive defense following the same principles would have been appreciated, but this could be more fitting for future works.
The content of the article is quite dense, even if detailed, and was hard to read.

**Questions:**

Have you tried training jointly the monitors using the same RL process, in the loop?

---

> ### Author Response · Authors · 2025-11-28
>
> We thank you for your positive assessment and for underscoring our systematic study. Below we address the concerns raised in the review:
>
> ---
> ## Regarding Adaptive defences
>
> We do not train the monitors jointly, as that would result in the RL training process having very high variance.
>
> We do, however, try to answer a similar question of whether probes trained on the model can detect a hidden backdoor during inference in Section 4.
>
> We also conduct an experiment to assess how easily the probe can *learn to detect* these new activations naively, as shown in Appendix N of our updated draft. We find that it can learn this with 5% of the data replaced with the backdoored policy.
>
> The most interesting way to use this model organism in the future would be to try and elicit these backdoor examples using auditing techniques [1], or detect them in an unsupervised way. Providing a sort of worst-case guarantee for both monitoring and whitebox auditing.
>
> We add this as a direction of future work in Section 6.
>
> ---
> ## Readability
>
> We address this in our shared response. We would love to hear any additional suggestions you might have!
>
> ---
>
> ## References:
>
> [1] Marks, S., Treutlein, J., Bricken, T., Lindsey, J., Marcus, J., Mishra-Sharma, S., Ziegler, D., Ameisen, E., Batson, J., Belonax, T. and Bowman, S.R., 2025. Auditing language models for hidden objectives. arXiv preprint arXiv:2503.10965.

---

### Official Review · Reviewer_YrVh · 2025-10-31

**Soundness:** 3
**Presentation:** 3
**Contribution:** 3
**Rating:** 6
**Confidence:** 2

**Summary:**

This paper investigates the problem that LLMs can intentionally evade latent-space level safety monitors. The authors propose RL-obfuscation, where model is rewarded from maintaining original output performance while minimizing the possibility of being detected by latent-space detectors. They found that token-level monitors are vulnerable while more holistic monitors remain robust. Experiments across multiple model sizes support their findings.

**Strengths:**

1. The problem is novel and interesting. This is an interesting red-team method showing that LLMs can be adversarial trained to bypass latent-space monitoring system.

2. The paper is well written and well presented.

3. The findings are insightful and the analysis are in-depth, demonstrating that different types of monitors shows different robustness level toward this attack method.

**Weaknesses:**

1. The paper would be beneficial to expand the scope of tasks. Current experiments focus on synthetic tasks. Adding more full instruction-following or open-domain generation tasks can strengthen the paper.

2. More interpretability on explaining how internal representations shift or which features the model manipulates can further strengthen the paper.

3. More discussion on key-components on more robust latent-space monitors can further strengthen the paper.

**Questions:**

Please address the weakness above.

---

> ### Author Response · Authors · 2025-11-28
>
> Thank you for your time and positive review. We address your concerns point-by-point below:
>
> ---
>
> ## Dataset used
>
> It is true that exploring more ambitious datasets would be interesting. However, we wanted reward signals with little ambiguity for this work. We have mentioned this as a part of our limitations and future work under Section 6 in our original draft.
>
> ---
>
> ## Interpretability
>
> Under section 5, Figure 5(b) describes the representation of which tokens are effected the most at the probing layer. We see that different types of texts are effected in different magnitudes, and it has some dependency on how often they occur in the generations.
>
> Our main Figure also shows a PCA plot, which shows that the new representations form a third cluster, distinct from benign and harmful generations.
>
> In addition to all of this we add a **new experiment in Appendix L**, trying to decompose which layers are responsible for the drop in recall.
>
> Are these points what you were looking for?
>
> We would also like to note that completely interpreting the model organism we introduce was not the main goal of the paper, but it was to test this capability of evading probes in LLMs and check if we get any real signal.
>
> ---
>
> ## Discussion on robust monitors
>
> We have addressed this point in our shared response, and would request referring to that.

---

### Author Response · Authors · 2025-11-28
**Shared Response**

Thank you all for your thoughtful feedback. We are pleased to learn that the reviewers found our study to be “insightful” and “well-presented”, and that our research direction is considered “important” and “addresses a critical gap in AI safety”. We address some common points and summarise key changes to our draft in this shared response.

---
## Readability

A common request in the reviews was to make the paper clearer and/or less dense, and to include more discussion about the monitors. We implore the reviewers to suggest any more concrete feedback they might have to improve its readability and summarise what we already changed in our draft below:

**Section 4 RQ2**
  - As Reviewer #fuAR mentioned, the results being spread across the main body and the appendix was unclear. We replaced Table 1 with a scatter plot between Evasion Rate and StrongReject Score.
  - Reviewers #YrVh, #fuAR and #z1Po requested more discussion on why some probes are more robust. We initially had this discussion in the Appendix and may have been missed by the readers. We moved this to the main body and added a new section in the Appendix to make the rest of the discussion more prominent in our updated draft (See Appendix F). Thank you for suggesting this.

**Rest of the main body**
  - As per Reviewer #EuSV’s recommendation to make the paper less dense, we make two changes in places which we believe had the most new information:
    - Figure 4(a) shows a pictorial describing Setup 2 (Backdoor Setup), and simplify the description of this setup in Section 3
    - Figure 5(a) has hatches displaying what each drop in recall across the bar plot means
  - As per Reviewer #fuAR’s request, we update our introduction to better describe our motivation.

---
## Key Experiments

We also summarise the key experiments added here. We add Appendix K to N and D.1, showing:
  - Generalizability of the probes and RL Obfuscation to jailbreak attack distributions (Appendix K)
  - Evaluating more activation monitors (Appendix L)
  - A more conclusive take on the change in activations (Appendix M)
  - Attributing how much each layer's attention/MLP contribute to the drop in probe performance (Appendix M)
  - How easily can the probe relearn the new policy (Appendix N)
  - Sensitivity to seeds (Appendix D)

---

### Meta-Review · Area_Chair_hbsK · 2026-01-13

**Summary:**

This paper studies whether LLMs can learn to evade latent-space safety monitors. The authors proposed RL-Obfuscation, which RL-finetunes a model to keep harmful output behavior while reducing detection by a fixed monitor. Results suggest token-level / mean-aggregation probes are easy to evade, while max-pooling and attention-based probes are harder to evade. There were four reviewers with two positive with low confidence and two negative with higher confidence. The authors did improve the presentation to make the paper more readable and added some experiments in the rebuttal but the AC thinks there are several concerns that are not well addressed.

**Reviewer Concerns:**

- **More realistic tasks**. The authors justified staying with synthetic.
- **Interpretability of representation shifts** is not well addressed by only adding plots and a decomposition analysis.
- **Training jointly the monitors**. This is declined by the authors since high variance. The authors then argued that they tried a similar way in Section 4 which basically did not answer this question in the rebuttal.
- **Justify reward design**. Authors cite precedent and mention tried variants. The rebuttal does not provide decisive ablation evidence.

**Reviewer Scores:**

In summary, the AC thinks the main issue is not about writing and presentation. It is not enough to adjust readability to convince the reviewers, especially with low score and high confidence to change their scores. Considering the insufficient rebuttal and two negative reviewers with high confidence, the AC decides to reject this paper.

---

### Decision · Program_Chairs · 2026-01-26

Reject